# Bats generate lower affinity but higher diversity antibody responses than those of mice, but pathogen-binding capacity increases if protein is restricted in their diet

Daniel E. Crowley [1,2]☉*, Caylee A. Falvo [1,2]☉, Evelyn Benson [2], Jodi Hedges [2], Mark Jutila [2], Shahrzad Ezzatpour [3], Hector C. Aguilar [3], Manuel Ruiz-Aravena [1], Wenjun Ma [4], Tony Schountz [5], Agnieszka Rynda-Apple [2‡], Raina K. Plowright [1,2‡]

1 Department of Public and Ecosystem Health, Cornell University, Ithaca, New York, United States of America, 2 Department of Microbiology and Cell Biology, Montana State University, Bozeman, Montana, United States of America, 3 Department of Microbiology and Immunology, College of Veterinary Medicine, Cornell University, Ithaca, New York, United States of America, 4 Department of Veterinary Pathobiology, College of Veterinary Medicine, and Department of Molecular Microbiology and Immunology, School of Medicine, University of Missouri, Columbia, Missouri, United States of America, 5 Center for Vector-borne Infectious Diseases, Department of Microbiology, Immunology, and Pathology Colorado State University, Fort Collins, Colorado, United States of America

☉ These authors contributed equally to this work.
‡ These authors are joint senior authors on this work.
* dc778@cornell.edu

## Abstract

Bats are reservoirs of many zoonotic viruses that are fatal in humans but do not cause disease in bats. Moreover, bats generate low neutralizing antibody titers in response to experimental viral infection, although more robust antibody responses have been observed in wild-caught bats during times of food stress. Here, we compared the antibody titers and B cell receptor (BCR) diversity of Jamaican fruit bats (*Artibeus jamaicensis*; JFBs) and BALB/c mice generated in response to T-dependent and T-independent antigens. We then manipulated the diet of JFBs and challenged them with H18N11 influenza A-like virus or a replication incompetent Nipah virus VSV (Nipah-riVSV). Under standard housing conditions, JFBs generated a lower avidity antibody response and possessed more BCR mRNA diversity compared to BALB/c mice. However, withholding protein from JFBs improved serum neutralization in response to Nipah-riVSV and improved serum antibody titers specific to H18 but reduced BCR mRNA diversity.

## Introduction

Bats host a high diversity of viruses including some that are highly virulent in other species [1–3]. Field, experimental, and modeling investigations have provided evidence of persistence of viral infections in individual bats [4–9]. These persistent viral infections suggest a unique host–pathogen interaction likely mediated by the host's immune response. It is proposed bats

**Data Availability Statement:** The data and code to recreate all figures are available here: https://zenodo.org/records/12825679.

**Funding:** This work was supported by National Science Foundation (Rules of Life scheme EF-2133763/EF-2231624 to ARA and RKP, Coupled Natural Human Systems DEB-1716698 to RKP), Defense Advanced Research Projects Agency (PREEMPT program Cooperative Agreement D18AC00031 to RKP, ARA, TS), National Institutes of Health (R01 AI134768 to WM & TS and R01 AI109022 to HCA). The content of the information does not necessarily reflect the position or the policy of the U.S. government, and no official endorsement should be inferred. Funders played no role in the study design, data collection and analysis, decision to publish, or preparation of the manuscript.

**Competing interests:** The authors have declared that no competing interests exist.

**Abbreviations:** AID, activation-induced cytidine; BCR, B cell receptor; IAV, influenza A virus; IP, intraperitoneal; JFB, Jamaican fruit bat; LLPC, long-lived plasma cell; MBC, memory B cell; MLN, mesenteric lymph node; SHM, somatic hypermutation; TF-IDF, term frequency-inverse document frequency; UMI, unique molecular identifier; VSV, vesicular stomatitis virus; WT, wild-type.

may generate a weaker-than-expected antibody response to pathogens and this could compromise their ability to clear infections, facilitating persistent infections [7,10].

When bats have been infected alongside other mammals, bats have generated the weaker antibody responses [11–13]. However, these comparative studies used viruses that are coevolved with the experimental bat species, so the weaker response seen in bats could be due to coevolutionary history with the virus. Single-species studies that have infected or immunized bats resulted in lower serum titers than researchers expected [14–16]; however, without a comparison species it is difficult to predict what antibody response should have been expected. It could be more feasible to draw conclusions about bats' antibody responses if bats were immunized alongside model organisms using well-studied antigens, such as T-dependent antigens which can have well-characterized responses in lab mice [17–21].

There are few proposed mechanisms in bats which explain their purported low antibody: antigen bond strength (i.e., antibody affinity). Antibody affinity typically increases after exposure to a foreign antigen. This process, known as affinity maturation, is mediated by the B cell-specific mechanism somatic hypermutation (SHM) and can be induced following infection and immunization with T-dependent antigens. Previous research hypothesized that the affinity maturation process is reduced in *Myotis* bats because the bats may have an expanded antibody variable region gene repertoire, limiting the need for SHM to generate a sufficient antibody repertoire. Subsequent studies found that following infection, *Artibeus* bats did not increase the expression of AID (activation-induced cytidine deaminase), a critical enzyme which is needed for SHM [22]. While either mechanism could reduce affinity maturation and lower antibody affinity, the evidence behind these claims is weak.

In contrast to the weak antibody responses observed in experiments, field sampling has occasionally identified wild bats with high titers of serum antibodies [23,24]. While it is challenging to establish causality in ecological settings, bats' body condition or food availability has been observed to correlate with pathogen-specific serum titers [24], seroprevalence [8,23], or total serum antibody (IgG) levels [25]. Furthermore, in some bat spillover systems, food availability is thought to be a key driver of pathogen shedding and spillover [26,27], but an immunological mechanism linking the 2 has not yet been established. Any prediction as to how antibodies might be impacted by food shortages is complex, as resource provisioning in wildlife systems, which might be assumed to enhance fitness, can correlate with decreasing serum antibodies [25,28–32]. Interestingly, experimental studies in mice have shown that manipulating metabolic pathways changes antibody affinity and germinal center dynamics, the primary location of SHM for B cells [33–35]. While many of these experiments relied on rapamycin or gene knockout mice to modify metabolic pathways, new work has demonstrated that diet manipulation alone changes the B cell receptor repertoire diversity in mice [36]. Together, these results suggest a complex interaction between food availability, nutritional status, and the antibody responses.

To directly test whether bats generate lower affinity antibodies than mice and to determine if diet may impact the affinity of their antibodies, we performed a series of 3 experiments on Jamaican fruit bats (*Artibeus jamaicensis*; JFBs). First, we immunized JFB bats alongside BALB/c mice with commonly used T-dependent or T-independent immunogens to measure their antibody response to a noninfectious challenge. T-dependent immunogens elicit SHM and affinity maturation, whereas T-independent immunogens result in an antibody response, but do not require T cell help and do not elicit SHM or affinity maturation. We hypothesized that bats would generate lower affinity antibodies than mice in response to T-dependent antigens but would generate comparable affinity antibodies to mice with a T cell-independent antigen. We next assessed how sensitive bats' antibody responses were to changes in their diet. We started by removing the protein supplement from the standard JFB diet, then challenging

the bats with a replication incompetent Nipah virus VSV (Nipah-riVSV). We then progressed to a viral infection study using the bat origin influenza A-like virus H18N11 [37,38]. We hypothesized that removing the protein supplement from their diet would have a deleterious effect on their antibody response.

## Results

### Bats develop delayed and reduced antibody responses to T-dependent antigens

We began by testing the hypothesis that bats would develop lower affinity antibodies than mice following immunization with a T.D. antigen but would have equivalent affinity antibodies to a T.I. antigen. We compared serum antibodies of mice and bats following intraperitoneal (IP) immunization and boosting with either 4-Hydroxy-3-nitrophenylacetyl conjugated to chicken gamma globulin (NP-CGG) (T-dependent (T.D.)) or 4-Hydroxy-3-nitrophenylacetic conjugated to Ficoll (NP-Ficoll) (T-independent (T.I.)) (Fig 1A).

We began our comparison study by using an indirect ELISA, one of the standard techniques to assess antibody titers. Using an indirect ELISA, mouse endpoint titers appeared to be higher than bats endpoint titers both post-immunization and post-boosting. This was true for both NP-CGG (TD) and NP-Ficoll (TI) (S1A and S1B Fig). However, for indirect ELISA results our endpoint titers for mice were equivalent to bats when we repeated the indirect ELISA but changed the secondary detection reagent from an anti-mouse IgG monoclonal antibody to protein G, a streptococcal protein which binds mammalian IgG and can be used when species specific anti IgG antibodies are not available (S1C Fig). This result demonstrated the endpoint titer value was a measure of both serum antibodies and the affinity of the secondary detection reagent. Furthermore, using protein G for both bats and mice could not eliminate the bias introduced by species specific reagents, as protein G has different affinities for different species [39]. Thus, we concluded the endpoint titer of indirect ELISAs was sensitive to the secondary reagent and could not be reliably used to compare the antibody responses of mice and bats.

To enable comparisons of mice and bats, we developed a competition ELISA for NP-CGG (TD) and NP-Ficoll (TI) (Figs 1D and S3A). Our competition ELISA showed that post boosting (day 35 and day 56), NP-CGG immunized mouse serum displaced the B1-8 competitor antibody while bat serum did not displace the B1-8 competitor antibody (day 35: DF 1, F value = 61.65, P(>F) 4.99e-05; day 56: DF 1, F value = 9.871, P(>F) 0.0138). Prior to boosting (day 21), neither the NP-CGG immunized bat nor mouse serum displaced the competitor B1-8 antibody, indicating boosting is required to generate antibodies which compete with the B1-8 antibody (Fig 1B). Sera from NP-Ficoll (T.I.) immunized bats and mice could not displace the B1-8 antibody on days 21 nor day 35, indicating SHM and affinity maturation are required to generate antibodies which compete with the B1-8 antibody (Fig 1C). Mouse from NP-Ficoll (T.I.) immunized mice could displace the competitor antibody on day 57. Unfortunately, sufficient serum was not available to run the competition ELISA for each blood draw time point.

### Bat splenic B cell mRNA contained more sequence diversity than mice

We sequenced the B cell receptor (BCR) heavy chain mRNA in bat and mice splenocytes to assess the B cell clonal diversity. Our objective was to better understand why the immunized and boosted mice possessed higher affinity serum antibodies. The BCR mRNA is the rearranged gene segment that encodes for the heavy chain of the antibody protein. Germinal center B cells undergo clonal bottlenecks as they mature. This clonal bottleneck reduces B cells

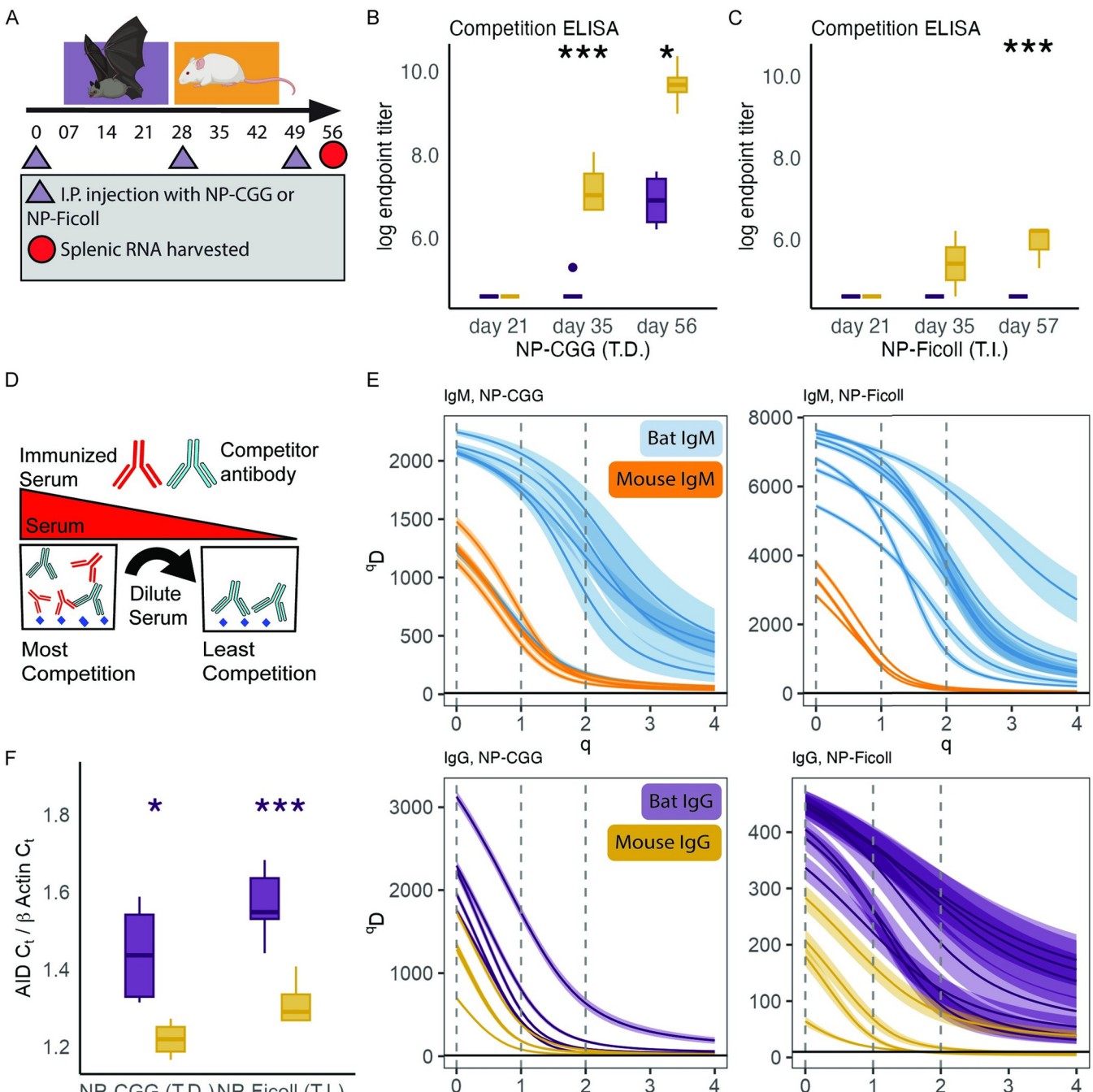

**Fig 1. Experiment 1; mouse and bat side by side experiment show bats develop low affinity antibodies and have more BCR diversity in their spleens.** (**A**) Timeline of immunization, boosting, bleeding, and tissue collection. (**B**) Competition ELISA endpoint titers from bats and mice immunized and boosted with NP-CGG (T-dependent). (**C**) Competition ELISA endpoint titers from bats and mice immunized and boosted with NP-Ficoll (T-independent). (**D**) Competition ELISA visual schematic. To avoid the bias of the secondary ELISA reagents, a competitor antibody was kept at a single concentration while immunized serum was added at increasing dilutions. As the immunized animal serum is increased, its antibodies can displace the competitor antibody if it has greater affinity. A more detailed schematic is shown in S3 Fig. (**E**) Splenic B cell receptor IgG (top panels) and IgM (bottom panels) mRNA diversity from bats and mice against NP-CGG (left panels) and NP-Ficoll (right panels). Higher $^qD$ values indicate greater diversity within the sample. When $q$ equals 0, the $^qD$ value reflects the total count of unique B cell receptor sequences within the data set. When $q$ equals 1, $^qD$ reflects the Shannon diversity measurement. When $q$ equals 2, $^qD$ reflects the Simpson Diversity measurement. Each line represents an individual animal, and a ribbon depicts the 95% CI, generated by bootstrapping. Overlapping ribbons indicate no significant differences between individuals. (**F**) Splenic AID mRNA CT levels normalized to beta actin for NP-CGG (right) and NP-Ficoll (left). Boxplots show the 25%, 50%, 75% percentiles, lines indicate the smallest and largest values within 1.5 times the interquartile range, dots indicate values beyond that. When an endpoint titer could not be estimated, the value was set to that of the pre-immunization serum. *P*

values derived from F statistic. Significance codes: "****"$P < 0.001$, "***"$P < 0.01$, "*"$P < 0.05$. All code and data to recreate figures can be found at https://zenodo.org/records/12825679. The mouse and bat images were modified from images sourced from BioRender.com.

diversity as cells with only the highest affinity BCRs are instructed to stay in the germinal center [40]. Our sequencing approach yielded approximately 17 million R1 and R2 reads. During the library preparation, a unique molecular identifier (UMI) was added to the 5′ end of cDNA strands. During sequence cleaning, reads that shared a UMI were collapsed into a consensus read. Prior to building a consensus sequence, we had approximately 180,000 reads per sample from the bat and mouse comparison study and 90,000 reads per sample from the food restriction study. After collapsing reads by UMI, we had approximately 59,000 reads per individual in the mouse and bat comparison study, but only 3,600 reads per individual in the food restriction study, but the average consensus count per sequence was approximately 7× higher. Of all reads, 61% had both a V, D, and J segment that was identifiable by IgBlast and was used in downstream analyses.

We assessed the BCR diversity in the spleen using Chao1 rarefied Hill's Diversity Curves [41–43]. Hill's Diversity uses input values (q) to generate a $^qD$ index of diversity; q ranges from 0 to infinity with interpretable outputs at specific values. When $q = 0$, $^qD$ is richness (the total number of unique BCR sequences), when $q = 1$, then $^qD$ is the Shannon Diversity measure, and when $q = 2$ then $^qD$ is a transform of the Simpson Diversity measurement. Simpsons Diversity is a measure of the probability that 2 randomly selected BCR sequences belong to the same BCR sequence.

At day 56, following immunization and boosting with both NP-CGG (T.D.) and NP-Ficoll (T.I.), the bat BCR IgM heavy chain diversity had increased sequence richness ($q = 0$), Shannon–Hill diversity ($q = 1$), and Simpson–Hill diversity ($q = 2$), compared to mouse BCR IgM heavy chain diversity (Fig 1E). We inferred this from the separation of 95% confidence intervals generated by bootstrapping.

At day 56, following immunization and boosting with both NP-CGG (T.D.) and NP-Ficoll (T.I.), the bat BCR IgG heavy chain diversity had increased sequence richness ($q = 0$) and Shannon–Hill diversity ($q = 1$) compared to mouse BCR IgG heavy chain diversity (Fig 1E). There was no clear separation of Simpson–Hill diversity ($q = 2$) by species. We inferred this from the separation of 95% confidence intervals generated by bootstrapping. Together, the sequence richness result indicates bats had more unique IgG heavy chain mRNA sequences, but the Simpson–Hill diversity result indicates the frequency of the most dominant BCR sequences were comparable between bats and mice.

We next compared AID mRNA levels in the spleens of mice and bats. AID is a critical enzyme for somatic hypermutation and class switching in B cells; however, previous studies found JFBs did not up-regulate AID in their spleens after infection with Tacaribe virus [22,44]. We found that bats immunized with NP-CGG (T.D.) increased AID expression over bats immunized with NP-Ficoll (T.I). However, bats had lower levels of AID expression than mice, regardless of T.D. or a T.I. immunization (Fig 1F).

## Removing dietary protein improved neutralization of Nipah-riVSV

We next tested the hypothesis that removal of dietary protein would have a detrimental impact on the antibody levels of bats (S4 Fig). We tested this hypothesis for 2 reasons. First, food availability is linked to both pathogen shedding, but also antibody levels, in several bat spillover systems. In the Australian Hendra virus system, *Pteropus* bats have been observed switching from their preferred high protein pollen diet to cultivated low protein fruits during winter months [45], the season in which Hendra virus spillovers occur [27] and starvation events in this

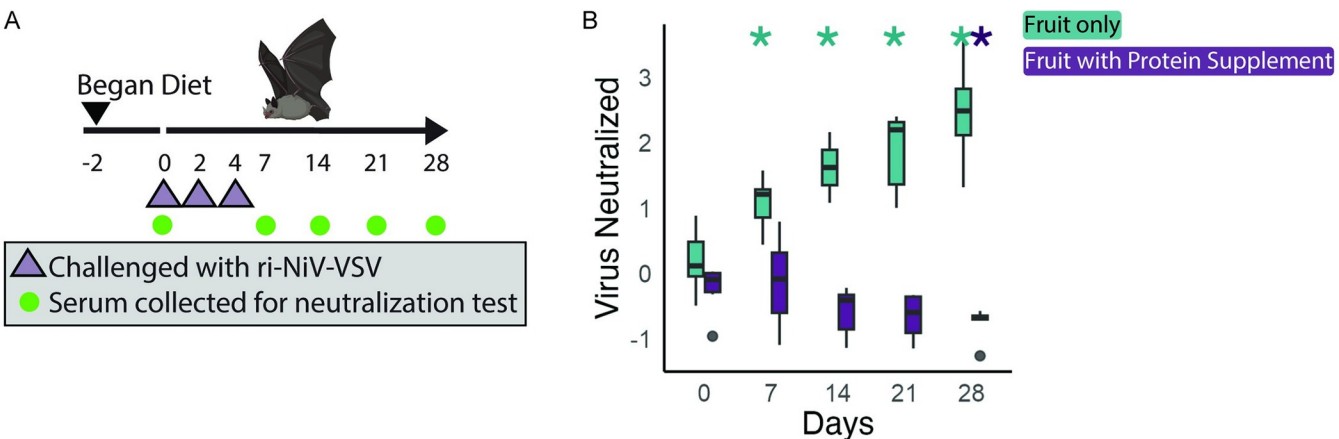

**Fig 2. Experiment 2; diet manipulation impact on bat humoral immune response to Nipah-riVSV challenge.** (**A**) Timeline of the experiment. (**B**) Serum neutralization of Nipah-riVSV entry. Y axis units refer to the reciprocal of Nipah-riVSV entry, normalized to day 0. Asterisks indicate significant difference compared to day 0 values, within a diet. Results are shown with serum at a 1:30 dilution. Boxplots show the 25%, 50%, 75% percentiles, lines indicate the smallest and largest values within 1.5 times the interquartile range, dots indicate values beyond that. *P* values refer to Student's *t* test ("*" *P* < 0.005 for multiple test correction). All code and data to recreate figures can be found at https://zenodo.org/records/12825679. The mouse and bat images were modified from images sourced from BioRender.com.

system have been correlated with increased seroprevalence in anti-Hendra IgG antibodies [23]. Second, housing conditions have known impacts on the immune system of laboratory animals reviewed in [46] and we were interested in assessing the effect of removing the protein supplement from the bats' diet.

We tested the impact of a low protein diet on antibody production in response to Nipah-riVSV (Fig 2A). Counter to our hypothesis, bats fed a fruit-only diet developed a neutralizing titer as early as day 7 (t = 4.74, *p* = 0.005, using a Bonferroni multiple test correction), while bats fed a protein supplemented diet did not develop a neutralizing titer (Fig 2B). Surprisingly, by day 28, bats fed a protein supplemented diet had sera that reduced viral neutralization relative to day 0, although the mechanism behind this observation is unclear (t = −9.47, *p* = 0.0002). Splenic mRNA was not harvested from these bats. After the neutralizing assay, no serum was remaining to use the F4 mAb antibody.

## Removing dietary protein improved antibody levels in response to H18N11 infection

We next assessed if a similar effect of protein restriction results could be replicated with the bat-specific influenza A-like virus (H18N11) (Fig 3A). We also started the bats on their diet for 3 weeks prior to viral infection to give the bats and period of adjustment to the diet prior to infection. We harvested RNA from mesenteric lymph nodes (MLNs) and spleen tissue to sequence B cell receptor mRNA.

Again, counter to our original hypothesis, at the terminal time point (day 20 post infection) the bats on the fruit-only diet had a stronger antibody response to the H18 antigen when we measured all antibody subclasses (i.e., IgG, IgA, IgM, etc.). Specifically, at the terminal time point the fruit-only diet had higher anti-H18 antibody levels when measured using the F4 monoclonal antibody (day 20: DF 1, F = 15.12, *p* = 0.003) (Fig 3D). The F4 mAb recognizes a 25 kDa protein in reduced and denatured S-300 size fractionated black flying fox (*Pteropus alecto*) sera (Fig 3B); 25 kDa is the approximate size of the of Ig light chain. Thus, measurements taken using the F4 mAb do not differentiate antibody subclasses. Interestingly, we

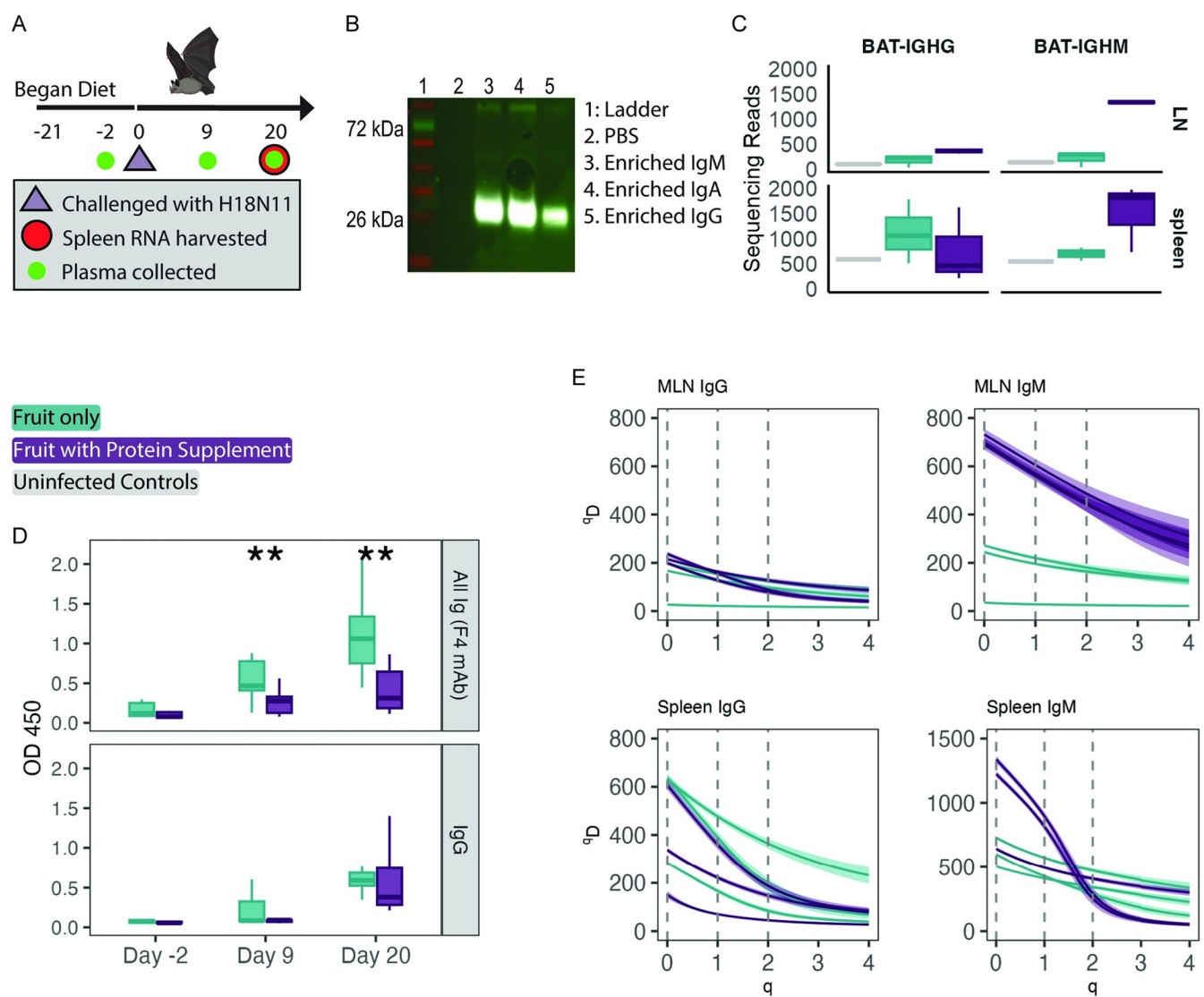

**Fig 3. Experiment 3; diet manipulation impact on bat humoral immune response to H18N11 infection.** (**A**) Timeline of the experiment. Plasma was collected instead of serum. (**B**) The F4 mAb tested by western blot against bat serum enriched by size for specific subclasses. (**C**) Counts of the cleaned BCR sequence reads by subclass (IgG left, IgM right) and tissue (mesenteric lymph nodes "LN" top, spleen bottom). (**D**) Plasma antibodies after infection with H18N11 (ELISA). Plasma was tested with protein G-HRP (bottom) or the F4 monoclonal antibody (top). Statistical tests are comparing the effect of diet within a day (i.e., OD450 on day 20 for bats on protein supplement versus fruit only diet). (**E**) Splenic B cell receptor IgG (left panels) and IgM (right panels) mRNA diversity from MLNs (top panels) and spleen (bottom panels) of bats. Negative controls are specifically omitted because of their low sequence counts as Chao1 rarefaction results in unreliable estimates when sample sizes are low. Higher $^qD$ values indicate greater diversity within the sample. When q equals 0, the $^qD$ value reflects the total count of unique B cell receptor sequences within the data set. When $q$ equals 1, $^qD$ reflects the Shannon diversity measurement. When q equals 2, $^qD$ reflects the Simpson diversity measurement. Each line represents an individual animal, and a ribbon depicts the 95% CI, generated by bootstrapping. Overlapping ribbons indicate no significant differences between individuals. Boxplots show the 25%, 50%, 75% percentiles, lines indicate the smallest and largest values within 1.5 times the interquartile range, dots indicate values beyond that. *P* values refer to Student's *t* test ("*"*P* < 0.05, "**"*P* < 0.005). All code and data to recreate figures can be found at https://zenodo.org/records/12825679. The mouse and bat images were modified from images sourced from BioRender.com.

observed no difference between diet groups in anti-H18 antibody levels when measured using protein G, which binds mammalian IgG (day 20: DF = 1, F = 3.2, P(>F) = 0.10; day 9: DF = 1, F = 2.57, P(>F) = 0.14) (Fig 3D). We did not have a neutralization assay established for the H18N11 virus and could not measure the antibodies' neutralization capabilities.

## BCR sequencing from H18N11 infected bats yielded few reads and diet-based shifts in diversity

We had observed in the comparative immunology experiment that bats had both lower affinity antibodies and higher BCR diversity. Thus, we were interested in assessing if the higher titers observed in the fruit-only diet would be associated with similar changes in BCR diversity. Because H18N11 was a G.I. pathogen, we collected abdominal draining MLNs mRNA in addition to splenic mRNA on day 20 post infection. Overall, we found fewer BCR sequencing reads in the MLNs than the spleen, with the notable exception of IgM sequences in the MLN of bats on the protein diet (Fig 3C). Read counts were greater in infected bats than in controls. We also examined the diversity metrics via the Hill diversity curves. We found few differences by diet group. Again, the notable exception of the diversity of IgM sequences in the MLN where bats on the protein diet had higher diversity in the sequence richness (q = 0), Shannon–Hill diversity (q = 1), and Simpson–Hill diversity (q = 2) compared to the spleen (Fig 3E). We arrived at these conclusions from the separation of the 95% CI generating by bootstrapping with replacement on Chao1 rarefied Hill diversity estimates. However, Hill diversity curves with Chao1 rarefication may have unreliable estimates when sample sizes are low or uneven [42,43], which is what we observed in the food restriction experiment, especially from MLNs harvested from bats on the fruit-only diet (Fig 3C). Thus, these Hill diversity estimates should be interpreted with caution and could be driven primarily by read depth.

## Discussion

Our results suggest that Jamaican fruit bats generate lower avidity serum antibodies compared to mice; however, avidity is improved when protein is removed from their diet. Specifically, we found that these bats generated a lower avidity antibody response to NP-CGG, a T-dependent antigen, compared to BALB/c mice. Our BCR mRNA sequencing results provide a candidate mechanism for this phenotype. Germinal center B cells undergo clonal bottlenecks as they mature [40], and we observed that the bats' lower antibody avidity coincided with higher BCR alpha diversity and lower expression of AID compared to mice. However, we also demonstrated that in response to either H18N11 influenza A virus (IAV) or Nipah-replication incompetent vesicular stomatitis virus (Nipah-riVSV), removing dietary protein from the bats' diet resulted in higher serum titers and decreased diversity of MLN and splenic B cell mRNA. These results suggest that the low antibody titer phenotype reported and observed in bats is sensitive to changes in their diet and provides context for the correlation between body condition, food availability, and seroprevalence observed in wild bats [8,23–25].

Our BCR sequencing data provides a candidate mechanism behind the low avidity antibody responses observed in JFBs that can be further explored in subsequent studies. By grouping BCR mRNA sequences of bats and mice into clones, we provide evidence that bats' B cell response is characterized by higher alpha diversity than BALB/c mice post immunization and boosting. There are several possible mechanisms which could explain the association of higher B cell clonal diversity and low avidity serum antibodies. Germinal center B cells undergo clonal bottlenecks when only B cells with high affinity BCR are instructed to stay in the germinal center, leading to a reduction in diversity [40]. Second, late-stage germinal center B cells differentiate into plasma cells [47], which generate far more BCR mRNA per cell than non-plasma cells [48]. If bat germinal centers are more permissive, enabling low affinity B cells to continue proliferating, or if plasma cell differentiation rates are lower in bats, this could result in the fewer dominant B cell clones and contribute to low avidity serum antibodies. However, more research will be needed to assess which processes are contributing. One important limitation is our sequences came from a single point in time and from unsorted splenocytes because there

is a lack of reagents for sorting bat lymphocytes. B cell training and selection within germinal centers is a dynamic process, and disentangling how bats' germinal centers mature over time will require the development of new methods and reagents specific for bat lymphocytes.

Our study cannot be used to make inferences into the rates of SHM in bats. SHM is critical for generating high-affinity B cells in mammals. A previous publication hypothesized some bat species' B cells may undergo SHM at rates lower than observed in model species because they possess an expanded number of V heavy chain ($V_H$). This in turn enables these bat species to generate a robust B cell germline repertoire from just VDJ rearrangement, hypothetically. However, other bat species appear to have a similar number of germline $V_H$ genes as humans [49,50]. Our study did not assess rates of SHM nor do our BCR mRNA diversity metrics provide reliable evidence of our bats' germline repertoire. Considering that SHM is evolutionarily conserved and is found in basal vertebrates [51], we suspect it is unlikely that bats have lost this process. Indeed, JFBs in our study and previous studies express AID [52], a critical enzyme for SHM. While SHM is likely critical for the proliferation and differentiation of B cells, without an annotated immunoglobulin heavy chain variable gene locus for JFBs, the data from our study cannot be used to make inferences into the rate of SHM in these bats.

While we found bats generated a relatively low avidity antibody response compared to mice, we also found bats' antibody responses could be improved by manipulating their diet. Specifically, restricting their dietary protein increased serum antibody levels to both a replication incompetent Nipah-G VSV and a live virus, H18N11 IAV. Similar findings to our study have been observed in wild animals, where, counterintuitively, improved access to food has been associated with decreased antibody levels [25,28–32]. In line with these findings, a meta-analysis of malnutrition in human children also shows complex and often surprising results [53]. Observational studies in children found that serum IgG levels are not depressed in mild or moderately malnourished children following vaccination and only decrease in severely malnourished children [53]. More surprisingly, while malnourished children have depressed levels of secreted IgA levels in mucosal surfaces, serum IgA levels increase [53]. Future work should determine which serum antibody subclasses in bats are impacted by dietary changes and if these differential antibody responses lead to changes in protection to reinfection.

While the mechanisms behind these diet and antibody associations in our study and others remain unclear, the field of mouse immunology has identified previously unappreciated levels of interconnectedness between B cell development and metabolic pathways. Specifically, mTORC1 associated metabolic pathways has been shown to regulate germinal center dynamics [34,35] and the manipulation of mTORC1 can change antibody affinity [33]. Future work should assess if diet manipulation impacted antibody responses via an mTORC1 mediated pathway. It is possible that bat metabolic pathways are uniquely susceptible to dietary changes. For example, it appears that frugivorous bats have unique adaptations to quickly utilize fruit-based meals to power flight while simultaneously relying on relatively small fat stores to power flight between meals [54]. Future work should assess if and how these adaptions for flight impact their immune response.

Furthermore, as we begin to manipulate the diet and housing conditions of bats, we also need to consider the housing conditions, and strains, of mice. In our study, we used the inbred BALB/c mouse strain. However, incorporating outbred mouse strains could better match the genetic diversity found in our study's bats. Mouse inbreeding has led to divergences in the VDJ genes between the commonly used BALB/c and C57BL/6 strains [55], a loss of germline diversity in the BALB/c light chain region [56,57], and changes to the B cell response to immunization [58]. We should also consider using pathogen naturalized mice (mice which have encountered various pathogens in their environment) in future studies, reviewed in [46]. These mice have different circulating lymphocyte populations and it would be useful to assess

if mice with prior microbial exposure changes their B cell receptor diversity following immunization. The incorporation of naturalized and outbred mice in future comparative immunology studies involving bats, wild or captive, could yield more relevant and insightful results.

Working with bats in laboratory settings introduces specific limitations. Sex and reproductive hormones have well documented and complex effects on immune function [59]. For our comparative immunology study, we used female bats and do not know if reproductive hormones impacted their antibody responses. Furthermore, while the JFB are a widely used model species for bat research, they are one of more than 1,400 species of bats [60], and it is important to not assume these traits are conserved across all bat species. Finally, we should assess additional antigens and adjuvants in these comparative immunology studies to determine the consistency of these results with alternative reagents.

If studies continue to find bats have lower avidity serum antibodies, we should address if and why this trait has been selected for. High affinity antibodies are typically produced by long-lived plasma cells (LLPCs), which are B cells that secrete antibodies with high affinity towards their respective antigens. However, LLPCs also have a diminished ability to respond to escape mutants. In contrast to LLPCs, memory B cells (MBCs) are thought to produce antibodies with a lower affinity toward their respective antigen but respond more effectively to escape mutants [61–63]. Investing in B cells that are better equipped to recognize escape mutants may be beneficial to bats because of their constant exposure to new microbes. Bats can live in large, dense roost structures with numerous co-roosting species which enable transmission between species [64,65]. Bats live especially long lives for their body size [66], and it is suggested bats host more zoonotic viruses than any other mammalian clade [67]. Therefore, it may be advantageous for bats to invest in broadly neutralizing antibodies. Thus, the lower affinity antibodies that this study and others have identified in bats [11–16] may be a by-product of the bat humoral immune system evolving under these unique circumstances.

## Methods

### Ethics statement

All care and procedures were in accordance with NIH, USDA, and the Guide for the Care and Use of Laboratory Animals (National Research Council, 2011). Animal protocols were reviewed and approved by the MSU Institutional Animal Care and Use Committee (IACUC) under protocol number 2021–174. MSU is accredited by the Association for Assessment and Accreditation of Laboratory Animal Care (AAALAC; accreditation no. 713).

### Animals

Jamaican fruit bats (*Artibeus jamaicensis*, JFB) were obtained from the specific pathogen-free breeding colony at Colorado State University. JFBs were fed a standard diet of mixed fruits with a protein supplement. Female wild-type (WT) BALB/c mice were purchased from Jackson Laboratories and maintained at the Montana State University Animal Resources Center under pathogen-free conditions. All mice used in this study were 6 to 8 weeks of age. A summary of the animal experiments is provided in S1 Table.

### Experimental timelines and groups

**Experiment 1: Mouse-JFB comparison.** We assigned a total of 12 female bats and 8 female BALB/c mice into 2 treatment groups: The first group (*n* = 6 bats, *n* = 4 mice) received NP-Ficoll and sterile saline i.p. (a type II T-independent (TI) antigen, NP-AECM-Ficoll, Santa Cruz Biotech sc-396292) with sterile saline (NP-Ficoll + SS) via intraperitoneal (i.p.) injection.

The second group ($n = 6$ bats, $n = 4$ mice) received NP-CGG (TD antigen, Santa Cruz Biotech sc-396209) and alum i.p. All bats were fed a diet of fruit (e.g., honeydew, cantaloupe, watermelon, bananas, strawberries, oranges) with a supplement of ground Mazuri Softbill Protein powder (Mazuri Exotic Animal Nutrition SKU 0053414) (S4 Fig).

The NP-CGG with alum immunogen was i.p. injected into bats with a target volume of 50 μl per bat at a concentration of 1 μg/μl. We boosted animals on days 21 and 48. We collected blood on days −5, 0, 7, 14, 21, 28, 35, 42, and 49 via the cephalic vein. Animals were euthanized on day 56. We determined that one bat in the NP-CGG group was pregnant at the time of the immunization experiments and was immunized subcutaneously to avoid harming the fetus. The pregnant bat was not included in analyses unless stated otherwise. All bats were fed a diet of fruit with a supplement of ground Mazuri Softbill Protein powder.

**Experiment 2: Food restriction with Nipah-riVSV.**   To assess if diet affects the antibody responses of JFBs, we assigned a total of 8 bats to one of 2 dietary regimes. The first group received a fruit-only diet ($n = 4$, 2 males and 2 females). The second group ($n = 4$, 2 males and 2 females) received the standard diet (e.g., honeydew, cantaloupe, watermelon, bananas, strawberries, orange with a supplement of ground Mazuri Softbill Protein powder). Approximately 2 tablespoons of Mazuri Softbill Protein supplement were added each day. A photo of the fruit with protein supplement is included in S4 Fig. Bats were provided food ad libitum and the amount of fruit in the diet was the same for each group.

JFBs were challenged with a Nipah glycoprotein—replication incompetent vesicular stomatitis virus pseudotyped viral particles (Nipah-riVSV), generated as previously described [68,69]. Briefly, human embryonic kidney (HEK239T) cells were transfected with several DNA mammalian expression plasmids encoding the Nipah fusion and glycoprotein proteins. Twenty-four hours post-transfection, the transfected cells were infected with vesicular stomatitis virus (VSV) virions lacking the VSV glycoprotein gene. The supernatant from this culture was collected 24 h later and pseudotyped VSV virions with Nipah virus surface proteins were then purified from the supernatant using a 20% sucrose cushion.

Animals were immunized with Nipah-riVSV. Nipah-riVSV was administered 25 μl per nostril containing a total of 100 μg of Nipah-riVSV. This was repeated every other day for a total of 3 inoculations (days 0, 2, and 4). We collected sera on days 0, 7, 14, 21, and 28.

**Experiment 3: Food restriction with H18N11 virus infection.**   We designed a second food manipulation experiment to assess the effects of diet on the immune response of bats and viral shedding. We used the H18N11 influenza A-like virus for which *Artibeus* bats are a reservoir [38,70]. We assigned 8 bats each to the 2 diet regimes described previously. The first group received the fruit-only/low protein diet ($n = 8$ males). The second group received the standard diet (fruit supplemented with Mazuri Softbill Protein powder) ($n = 8$ males). Approximately 2 tablespoons of Mazuri Softbill Protein supplement were added each day. Only powder that stuck to the fruit was accessible to bats. Bats were provided food ad libitum and the amount of fruit in the diet was the same for each group.

Bats were placed on their diets for 3 weeks. After 3 weeks, 6 bats in each group were inoculated intranasally with $5 \times 10^5$ TCID$_{50}$ of H18N11 virus. Two bats per diet group were not infected and kept in a separate room. Bats were kept on their dietary regimes throughout the infection. Blood samples were collected, as previously described, on days −2, 3, 9, 15, and 20 (the final time point). After euthanizing all animals, spleens and MLNs were harvested for RNA. Three individuals per infection group were selected randomly for BCR sequencing. One negative control animal from each diet (noninfected) was also selected for BCR sequencing.

Briefly, the H18N11 influenza A virus used (A/flat-faced bat/Peru/033/2010) was rescued by using 8 plasmids reverse genetic system as described previously [71]. The rescued H18N11 virus was propagated in RIE1495 cells and titers determined on in MDCK II cells [37,72].

### Indirect ELISA assays

Antibody titers were assessed for all blood samples from Experiments 1, 2, and 4 as described in S1 Methods.

### Competition ELISA

We designed a competition ELISA to compare the antibody responses between JFBs and mice (Figs 1D and S3) as described in S1 Methods. This competition ELISA was designed to report an endpoint titer that was unbiased by secondary reagents for mouse and bat antibodies (S1C Fig) [39].

### B cell receptor (BCR) sequencing

B cell receptor library preps were generated from unsorted spleen and MLN cells from JFBs and mice, as described in S1 Methods. For BCR library preparation, we followed previous protocols, interchanging JFB IgM and IgG primers when appropriate [73].

### Bioinformatics steps

We processed, filtered, and analyzed the sequencing BCR reads using pRESTO and previously described germline free identification methods, as described in S1 Methods [74,75]. For processing BCR reads, we used mouse V, D, and J germline sequences, obtained from the international ImMunoGeneTics/GENE database (IMGT/GENE-DB) [76] on 2022-03-20. Bat IgG and IgM sequences were separated using pRESTO by primers listed in S2A Fig. Briefly, the TF-IDF (term frequency, inverse document frequency) quantified the distance between BCR sequences. We tested the number of nucleotides from the $3'$ end of the BCR sequence to use in analyzing sequences (Fig 2B). We used hierarchical Bayesian mixture models in Rstan 2.21.8 [77] to bin sequences into cluster ("clones"). We assessed clonal diversity between groups using Hill Diversity Curves and Chao1 rarefication methods to account for individual variation in read depth, implemented using the Alakazam package (version 1.2.1) [41–43,78,79] (Fig 2C). Statistical significance of the Hill Diversity Curves was assessed by bootstrapping the replacement the Chao1 rarefied Hill diversity estimates, as described previously [78].

We assessed how many nucleotides should be included in the BCR mRNA sequences to establish clones. Our analyses suggested including more than 110 nucleotides had marginal effects on downstream inferences (S2B Fig). Both the hierarchical and non-hierarchical Bayesian mixture models suggested a ~0.25 term frequency-inverse document frequency (TF-IDF) distance cutoff for categorizing sequences (S2C Fig). All sequence diversity inferences were made using 110 nucleotides from the constant region of the BCR. This cutoff included the VDJ junction, which accounts for most of the diversity in BCR sequences (S2A Fig) [44].

### Pseudotyped virus neutralization assay

We assessed serum neutralization of Nipah Virus F/G pseudotyped VSV particles from Nipah-riVSV virion-inoculated bats as described in the S1 Methods.

### Monoclonal antibody production

Hybridoma-derived monoclonal antibodies were generated using BALB/c splenocytes and SP2/0 myeloma cells. Mice were immunized and boosted with size fractionated *Pteropus* bat serum, as described in S1 Methods.

## Supporting information

**S1 Fig. Bias of indirect ELISA in species comparison data.** (A) Indirect ELISA endpoint titers from bats and mice immunized and boosted with NP-CGG (T-dependent). (B) Indirect ELISA endpoint titers from bats and mice immunized and boosted with NP-Ficoll (T-independent). (C) Indirect ELISA dilution curve from bats and mice immunized and boosted with NP-CGG. Serum is from day 56. Bat serum IgG was measured using protein-G as the secondary reagent. Mouse serum IgG was measured with either protein-G or a monoclonal antibody specific to mouse IgG as the secondary reagent. The dashed line at 0.5 indicates a theoretical OD450 cutoff to establish an endpoint titer. Serum dilution is shown on a natural scale. Box-plots show the 25%, 50%, 75% percentiles, lines indicate the smallest and largest values within 1.5 times the interquartile range, dots indicate values beyond that. When an endpoint titer could not be estimated, the value was set to that of the pre-immunization serum. *P* values derived from F statistic. Significance codes: "***"$P < 0.001$, "**"$P < 0.01$, "*"$P < 0.05$. All code and data to recreate figures can be found at https://zenodo.org/records/12825679.
(DOCX)

**S2 Fig.** (**A**) Graphical representation of BCR mRNA. Shown is the VDJ region, where much of the BCR diversity occurs. Also shown is the Illumina adaptor and the location where the species and Ig class-specific primers bound. The "Cutoff Distance for Establishing Clones" is a graphical representation of how many nucleotides were used to build our diversity models. (**B**) We assessed a range of cutoff values to build our diversity models. Before modeling diversity, we needed to classify mRNA BCR transcripts into clones. Without an annotated V, D, and J germline for Artibeus bats, we classified BCR transcripts into clones using the TF-IDF distance metric. We optimized the number of nucleotides to include when calculating the TF-IDF metric. We refer to the number of nucleotides as the "Cutoff Distance for Establishing Clones." For the optimization, we repeatedly subsampled sequences from our full BCR sequence library. For each sample, we trimmed the 5′ end of the sequence. After trimming the subsample, we clustered sequences into clones and calculated the percentage of sequences that classified as clones. When the "Cutoff Distance for Establishing Clones" was <50 nucleotides, the BCR sequence primarily consisted of the constant region and some of the FWR4 region. Since there is no variability in the constant region of the BCR sequence, almost all sequences were classified as a single clone. As the cutoff distance approached 110 nucleotides, the trimmed sequences encapsulated to VDJ region, which is the most variable region of the BCR sequence. (**C**) Histogram of the TF-IDF distance metric used to cluster BCR sequences. TF-IDF was used to compare sequence similarity. To establish a TF-IDF cutoff, we used a Bayesian mixture models to identify the TF-IDF cutoff that could cluster a group of sequences as a set of "clones." The posteriors of a Bayesian hierarchical mixture model are shown in red and the results from a nonhierarchical mixture model are shown in blue. Sequences shown here are from spleens of mice and bats immunized with NP-CGG and NP-Ficoll. Sequences were trimmed using 110 nucleotides. Boxplots show the 25%, 50%, 75% percentiles, lines indicate the smallest and largest values within 1.5 times the interquartile range, dots indicate values beyond that. The mouse and bat images were modified from images sourced from BioRender.com.
(DOCX)

**S3 Fig.** (**A**) An in-depth visual of the competition ELISA. Data shown comes from animals immunized with NP-CGG. The serum is from day 56. (**B**) As immunized animal serum is diluted out the OD450 increases. OD450 is indicating the amount of tagged competitor antibody that is binding. This approach is an attempt to remove the bias introduced by species-

specific reagents and the bias introduced by protein-G's different affinities for different species. All code and data to recreate figures can be found at https://zenodo.org/records/12825679. (DOCX)

**S4 Fig. A photo taken during the experiment of a representative meal given to bats within a cage.** The fruits in this photo are coated in the ground Mazuri Softbill Protein. (DOCX)

**S1 Table. Summary of experiments with sample sizes.** (DOCX)

**S1 Methods. Supplemental methods.** (DOCX)

**S1 Raw Images. Uncropped image of the western blot in Fig 3B.** (PDF)

## Acknowledgments

We thank Chris Grant for assistance in generating monoclonals; Armin Scheben analyzing the IgHV loci, Hannah Frank for advice on 5' RACE; Abby Luu and Monica Hall for assistance with experiments, Kerri Jones, Susan Carroll, Kirk Lubick, Ryan Barlett, Janet Baer, Lauren Cantamessa, Miles Eckley, Tracy Dolan for bat husbandry logistics; and Helen Dooley for help with assay design. The mouse and bat images were modified from images sourced from BioRender.com.

## Author Contributions

**Conceptualization:** Daniel E. Crowley, Caylee A. Falvo, Mark Jutila, Manuel Ruiz-Aravena, Tony Schountz, Agnieszka Rynda-Apple, Raina K. Plowright.

**Data curation:** Daniel E. Crowley, Caylee A. Falvo, Evelyn Benson, Jodi Hedges, Mark Jutila, Shahrzad Ezzatpour.

**Formal analysis:** Daniel E. Crowley, Raina K. Plowright.

**Funding acquisition:** Hector C. Aguilar, Tony Schountz, Agnieszka Rynda-Apple, Raina K. Plowright.

**Investigation:** Daniel E. Crowley, Caylee A. Falvo, Evelyn Benson, Shahrzad Ezzatpour, Tony Schountz, Agnieszka Rynda-Apple.

**Methodology:** Daniel E. Crowley, Agnieszka Rynda-Apple.

**Project administration:** Tony Schountz, Agnieszka Rynda-Apple, Raina K. Plowright.

**Resources:** Daniel E. Crowley, Hector C. Aguilar, Wenjun Ma, Tony Schountz, Agnieszka Rynda-Apple, Raina K. Plowright.

**Software:** Daniel E. Crowley.

**Supervision:** Daniel E. Crowley, Agnieszka Rynda-Apple, Raina K. Plowright.

**Validation:** Daniel E. Crowley, Evelyn Benson.

**Visualization:** Daniel E. Crowley, Manuel Ruiz-Aravena, Agnieszka Rynda-Apple, Raina K. Plowright.

**Writing – original draft:** Daniel E. Crowley.

**Writing – review & editing:** Daniel E. Crowley, Caylee A. Falvo, Evelyn Benson, Jodi Hedges, Mark Jutila, Shahrzad Ezzatpour, Hector C. Aguilar, Manuel Ruiz-Aravena, Wenjun Ma, Tony Schountz, Agnieszka Rynda-Apple, Raina K. Plowright.

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
