## [Editor Report · Decision Letter 0]

16 Jan 2024

Dear Dr Crowley, 

Thank you for submitting your manuscript entitled "Bats generate lower affinity, but higher diversity antibody responses compared to mice, an effect that can be manipulated with diet" for consideration as a Research Article by PLOS Biology.

Your manuscript has now been evaluated by the PLOS Biology editorial staff, as well as by an academic editor with relevant expertise, and I am writing to let you know that we would like to send your submission out for external peer review. After discussions with the Academic Editor, we have decided to seek re-review with a couple of additional reviewers who would help assess the previous reviews and assess the wild animal immunity and the and a B cell specialist. Since PNAS does not disclose the identities of their reviewers, we would seek the advice of new reviewers. We will provide them with the previous reports and rebuttal and ask them to assess the revisions made and continue the process from there to avoid having to start from scratch.

IMPORTANT: Your paper is submitted as a regular Research Article. However, after discussion with the Academic Editor, we think that your study would be better considered as a Short Report (https://journals.plos.org/plosbiology/s/what-we-publish#loc-short-reports). Very little re-formatting is required, but we would need you to reduce the number of Figure down to 4. You could do this either by combining multiple Figures or by moving some of the material in the existing main Figures to the supplement. Please do this, and select "Short Reports" as the article type, when uploading your additional metadata (see next paragraph).

Once your full submission is complete, your paper will undergo a series of checks in preparation for peer review. After your manuscript has passed the checks it will be sent out for review. To provide the metadata for your submission, please Login to Editorial Manager (https://www.editorialmanager.com/pbiology) within two working days, i.e. by Jan 18 2024 11:59PM.

Kind regards,

Melissa

Melissa Vazquez Hernandez, Ph.D.

Associate Editor

PLOS Biology

---

## [Decision Letter · Decision Letter 1]

13 Mar 2024

Dear Dr Crowley,

Thank you for your patience while your manuscript "Bats generate lower affinity, but higher diversity antibody responses compared to mice, an effect that can be manipulated with diet" was peer-reviewed at PLOS Biology. It has now been evaluated by the PLOS Biology editors, an Academic Editor with relevant expertise, and by two independent reviewers. 

In light of the reviews, which you will find at the end of this email, we would like to invite you to revise the work to thoroughly address the reviewers' reports. As you will see below, all reviewers are quite positive and interested in the work but agree that an important change in the study is necessary as well as some additional experiments. Both reviewers have concerns over the selection of mice model and the further comparison. Additionally, Reviewer #1 mentions that the claim regarding BCR diversity should be better supported. Reviewer #2 wonders what are the changes between T-dependent and independent responses with diet. Finally, the reviewers consider that the study should be more structured and connected. We agree with all reviewer concerns and would require experimental revisions to address them, as we consider that this would strengthen the work.

IMPORTANT: after discussion with the Academic Editor and the reviewers, given the significant concerns regarding the validity of comparison between the animal species, we strongly suggest for this part to be removed, which would not affect the impact to the study and would help in the flow of the writing. Additionally, the Academic Editor has made some suggestions, which you can find at the foot of this e-mail, to follow the reviewers comments and to improve the study. Also, keep in mind that Short Reports should contain no more than 4 figures in the main text. 

Given the extent of revision needed, we cannot make a decision about publication until we have seen the revised manuscript and your response to the reviewers' comments. Your revised manuscript may be sent for further evaluation by all or a subset of the reviewers.

We expect to receive your revised manuscript within 3 months, however please let us know if you would require some more time for revision, which would not be a problem. Please email us (plosbiology@plos.org) if you have any questions or concerns, or would like to request an extension.

**IMPORTANT - SUBMITTING YOUR REVISION**

*Re-submission Checklist*

*Published Peer Review*

*PLOS Data Policy*

*Blot and Gel Data Policy*

Sincerely,

Melissa

Melissa Vazquez Hernandez, Ph.D.

Associate Editor

PLOS Biology

REVIEWERS' COMMENTS

Reviewer #1: 

After reviewing the latest manuscript and considering two previous rounds of reviews, my overall opinion is that this paper presents highly intriguing results that could be suitable for a short paper. I concur with the earlier reviews that this is manuscript covers a substantial amount of work and commendable efforts to address the inherent difficulties that arise with comparative immunology in non-model species.

Were the authors to consider shortening this manuscript, this could be achieved by focussing on the main strength of the paper, which I believe are the challenge experiments under dietary manipulation (Figs 2, 3 and associated text). I fully support the need for a comparative approach and commend the efforts made to address biases caused by species-specific reagents, specifically, the competition ELISA. In addition, I believe the diversity metrics are interesting even if it isn't yet clear how they relate However I have serious concerns about other elements of this comparison. 

1. The choice of the BALB/c mouse as representative of "other mammals" (L69) as serious limitations. Indeed, unlike the Jamaican Fruit Bats used in this study, BALB/c mice are inbred, and thus homozygous at each allele, including the MHC. While generation of BCR diversity is only partly dependent on the genetic background, the lower BCR diversity in mice reported in figure 1H therefore could simply be a consequence of not having used an outbred mouse. As a consequence, statements like "Overall, JFBs generated a weaker antibody response and possessed more BCR mRNA diversity compared to mice" (L24-25) should probably be qualified, since these differences may be due to the study design rather than to biological characteristics of bats. The paper would be more compelling if the authors could replicate the BCR diversity study in outbred Mus musculus. 

2. It isn't clear how differences in BCR diversity impact antibody affinity. In particular, the claim that "affinity maturation process is reduced in Myotis bats due to an expanded antibody variable region gene repertoire [...]" (L63) is highly speculative, and would require experimental validation or further support from the literature. Of the citations given (Bratsch 2011, Buter 2011, Shaw 2012), only Bratsch mentions this idea but as far as I can tell, merely as a hypothesis. In fact, a diverse initial repertoire generated by V(D)J recombination can be beneficial because it increases the likelihood that at least some BCRs will have an initial, albeit low, affinity for a given antigen, providing a better starting point for subsequent affinity maturation. The authors should consider dropping this claim if a trade-off between affinity maturation and antibody variable gene repertoire cannot be better supported.

3. As a consequence of the previous point, it isn't clear how the JFB vs BALB/c comparison links up with the section on the effects of dietary protein, (experiments 3 & 4).

Based on the issues mentioned above, I would suggest completely removing the mouse comparisons from this paper, and better justifying the causal link or at least the complementarity between BCR diversity and antibody affinity. Additionally, there are a few minor points that should be addressed.

4. There should be better parity between the assays in experiments of Fig 2 (Nipah-riVSV) and Fig 3 (H18N11), especially analysis of the antibody subclasses and protective responses (neutralization or similar). Indeed, it is important to avoid readers assuming an antibody response is always protective.

3. The section "Macroscopic Findings of Lymphoid Tissue Responses" (L142-144) could be cut, as this has no straightforward implication for (protective) immunity. It might be sufficient to explain in the methods that enlarged LNs were extracted for further analysis.

5. In support of the author's view, and in light of the observation that there was no difference in IgG concentrations in response to H18N11 (L157), it would be good to discuss the apparent lack of protection conferred by IgG.

6. Supplemental Figure 1 is confusing. The legend implies that the plots represent both bats and mice, but it appears that only bats are depicted in panels C and D.

Reviewer #2: 

This review was completed by a PI and their postdoctoral fellow together with the permission of the editor. It reflects the feedback of both researchers. The paper by Crowley et al. presents some intriguing initial experiments on the humoral immune response of Jamaican fruit bats compared to the response of BALB/c mice, as well as the impact of diet on the humoral response. While preliminary, the results could be quite exciting. On balance, the discussion is well thought out, presents an interesting model for the adaptive significance of diverse but less neutralizing immunoglobulin populations, and full of appropriate caveats. However, we have several questions and concerns with the methodology and presentation of results that would clarify the strength of their results. If the major concerns and questions can be addressed and resolved, and the results still hold, it would likely make a great short report for PLoS Biology. We summarize our major concerns first and then suggest some minor, mostly text corrections. 

Major revisions:

The way the manuscript is currently structured is somewhat disjointed and difficult to follow. It reads as four pilot experiments loosely grouped into two stories. 

The results section can be hard to follow at times and reads more like a methods section in places - this could be remedied by including more rationale/justification/clarification for experimental decisions and the corresponding results.

The first story compares immune responses in bats and mice to non-infectious immune challenges. The second story looks at the effect of diet on antibody responses in bats to pathogens. Each experiment was conducted differently, with variation in the antigens used for challenges, challenge timings, experimental procedures (e.g. inoculation route; ELISA vs neutralization assays), analyses (some have BCR analyses and others do not), and study populations (all male; all female). This makes it quite hard to compare across studies - indeed are the studies comparable? -- and uncover general themes. The experiments are each presented separately in the results which exacerbates the piecemeal nature of the manuscript. In particular, the pilot study was conducted with different reagents and different timepoints, and it's unclear why they included this information in the manuscript. Can the authors provide more rationale for the pilot study and relate it better to the following studies? None of the results are directly comparable but perhaps the results could at least explicitly group lines of evidence that support similar hypotheses, e.g. there may be increased titers of antibodies in fruit-only groups? 

We would also like additional information to evaluate the validity of their results, specifically their calculations of BCR diversity. How are read counts quantified? What's the pipeline/normalization procedure? How are the authors accounting for variability in sequencing between samples? This is information that should be in the results section because it has huge impacts on the trustworthiness of the results. A visual inspection of figure 3c (the only place read depth is supplied) suggests a very strong correlation between sequencing depth and inferred diversity. Also why were the uninfected control data not plotted? The authors state that they used bootstrapping to create 95% confidence intervals and determine differences in diversity but don't explain how their bootstrapping was done. Looking in the documentation for Change-O it merely states it uses bootstrapping but many R packages that do bootstrapping do so by resampling with replacement to the original sample size. In this way the read depth could dramatically skew the results. The authors should subsample each individual down to the lowest read count of any sample and rerun their analyses, as well as examine how their inferred diversity metrics correlate with the read depth from each individual to ensure this is not an artifact. The authors also need to report their read counts and normalization for all BCR analyses, not just the flu infection experiment. 

We found the presentation of the ELISA results for the mouse-bat comparison confusing. Do both ELISAs need to be in the main figure? If the indirect ELISA can't be used to compare the antibody responses of bats and mice then I believe it would be better included as a separate figure, where the authors highlight the implications this has on other studies (in the vein of their response to a previous reviewer comment "We think this is an important point for readers, because we often see reports of 'low titers' in bats, but if this conclusion comes from an indirect ELISA, then this is a mistake." We agree this is important to highlight but this point is currently obscured and should be plainly stated.)

Also, we wonder about the implications of their indirect ELISA results versus competition ELISA results especially in comparison to their pilot study. The authors highlight that in the pilot they did not see increased titers in response to the T dependent antigen until after boosting but the indirect ELISA results (even if comparisons between bats and mice are inappropriate, within species between time point comparisons should be valid) shows an increase in titers without boost. 

Why are the analyses done for the H18 and Nipah infections different? The authors went into much more detail for the H18 infection - can these same analyses on the BCR sequencing diversity be done for Nipah? If not why? If not, then they should consider reworking the results section to make Nipah a smaller part and more of a supporting piece of data for observations from H18.

Lines 123-141, Figure 1H,I: In this study the authors used BALB/c mice as a comparison. BALB/c is a highly inbred lab strain of mouse that is known to be much more germline-focused, with less nucleotide addition in the CDR3 region than the human repertoire. Additionally, its V gene repertoire and immunoglobulin repertoire in general is very distinct even from C57BL/6, another inbred strain of Mus musculus (Collins et al. 2015, Phil Trans B, https://doi.org/10.1098/rstb.2014.0236). This known impact of inbreeding with likely lower diversity BCR repertoires would seem to bias the comparison between (relatively outbred) bats and a highly inbred, homozygous mouse line. In any case, the authors need to address how their choice of mouse impacts their comparison.

Line 174/ Figure 1E and F: Are the responses really "weaker" when compared to mice? They've shown more diversity in sequences even though there is less binding in a competition assay. What about straight B cell/antibody number? The response may be less specific, but not "weak" in terms of response rate/volume/activation. I have issues with the word "weaker" here. It could also be equally strong just not binding the exact same epitope that the competitor antibody binds and therefore not excluding it. Similarly, why were the competition assay results only done at three time points versus the 10 timepoints at which the indirect ELISA was performed?

How do the T-dependent and T-independent responses change with diet? Why was this experiment not done? Can the authors better link the initial mouse vs. bat study to the later diet studies?

The bioinformatics section of the methods needs to be expanded. The authors need to include program names and versions for all steps in the processing pipeline. In many places they state that things were done "as described previously," but there needs to be more explanation with a mention of the tools used and basic processing pipeline (which seems to be the immcantation portal?). Genome versions and citations (including the accession number) that were used for the BCR-seq processing should also be included. Additionally, if not already included in the submission, the authors need to include a supplemental table with the BCR sequencing metrics for all samples: at minimum this should include total reads sequenced, the number of mapped/aligned reads, the mapping/alignment rate, the number of reads used in the downstream analysis. This will also help a great deal with assessing the validity of their results.

Minor revisions:

There are some grammatical/spelling errors throughout the manuscript that need to addressed, including:

Line 52: "responses than were" should be "responses that were"

 Line 72: "elicits" should be "elicit"

Line 123: "germinal centers B cells" should be "germinal center B cells"

Line 576: The FWR4 region is not part of the C region but is part of the V region located on the 5' end of the C region. This should be corrected in the last sentence. 

Figure 2: y-axis needs units, if it's fold change, then I recommend stating that in the axis label for clarity

ACADEMIC EDITOR'S COMMENTS.

The reviewers appreciated the magnitude of the work done and saw the experimental test of effects of nutrition on bat immunity as a milestone for the field which if verified, merits publication. However, both reviewers raised serious concerns about the validity of comparing JFBs and BALB/c mice and it was recommended that this component be removed from the mansucript. I tend to support this suggestion. I add that focusing on the effects of diet on bat immunity would resolve some of the concerns from earlier reviewers about overgeneralizing the “bat-other mammal” comparisons, would make for a more compelling justification for the manuscript than the current emphasis on persistent viruses (which the studied viruses may not be), and would reduce the “disjointed” nature of the manuscript noted by both reviewers. The reviewers also flagged technical and/or data presentation issues which currently make it difficult to compare the experiments on H18 and Nipah and to evaluate the robustness of the intriguing effects on BCR diversity. If the manuscript can be re-focused and the core results are shown to be robust, this will be valuable contribution.

---

## [Decision Letter · Decision Letter 2]

12 Jul 2024

Dear Dr Crowley,

Thank you for your patience while we considered your revised manuscript "Bats generate lower affinity, but higher diversity antibody responses compared to mice, an effect that can be manipulated with diet" for consideration as a Short Reports at PLOS Biology. Your revised study has now been evaluated by the PLOS Biology editors, the Academic Editor and the original reviewers. 

In light of the reviews, which you will find at the end of this email, we would like to invite you to revise the work to thoroughly address the reviewers' reports. As you can see below, the reviewers appreciate the effort made during the revision. However, after discussion with the Academic Editor, two issues need to be addressed before the study can be accepted for publication. In line with Reviewer #1, we suggest expanding the discussion on the specific risks of using inbred mice. This will provide a more compelling rationale for future experiments with outbred mice. Additionally, the technical issue highlighted by Reviewer #2 regarding sequencing depth must be addressed by following the reviewer's suggestions. While we think the manuscript is moving in the right direction and we and the reviewers are clearly interested in it, this technical concern is important and should be addressed.

Please also attend to the following editorial concerns:

a) During re-submission please indicate in the Financial Disclosure option the same statement you have within the manuscript.

b) The Ethics statement should include the full name of the IACUC/ethics committee that reviewed and approved the animal care and use (which you have), as well as the protocol/permit/project license number (which is missing). Please include the specific protocol adhered to your ethics committee.

https://journals.plos.org/plosbiology/s/ethical-publishing-practice

Please supply the numerical values either in the a supplementary file or as a permanent DOI’d deposition for the following figures:

Figure 1BCEF, 2B, 3CDE, S1ABC, S2BC, S3B

d) Please cite the location of the data clearly in all relevant main and supplementary Figure legends, e.g. “The data underlying this Figure can be found in S1 Data” or “The data underlying this Figure can be found in https://doi.org/10.5281/zenodo.XXXXX”

e) We require the original, uncropped and minimally adjusted images supporting all blot and gel results reported in the Figure 3B

Please carefully read our guidelines for how to prepare and upload this data: https://journals.plos.org/plosbiology/s/figures#loc-blot-and-gel-reporting-requirements

We expect to receive your revised manuscript within 2 months. Please email us (plosbiology@plos.org) if you have any questions or concerns, or would like to request an extension. 

**IMPORTANT - SUBMITTING YOUR REVISION**

*Resubmission Checklist*

*Published Peer Review*

*PLOS Data Policy*

*Blot and Gel Data Policy*

Sincerely,

Melissa

Melissa Vazquez Hernandez, Ph.D.

Associate Editor

PLOS Biology

REVIEWERS' COMMENTS:

Reviewer #1: 

Thank you for carefully addressing the points I raised in my comments. The manuscript now holds together much better. While I'm not fully convinced that all the pieces fit seamlessly, I believe the rationale and conclusions drawn from the data are sufficiently argued.

Reviewer #2: 

The authors have done a great job of responding to the numerous reviewer comments. Let me say first that I appreciate how Herculean this effort is and the challenge of assessing humoral immunity in non-model species. In particular, I think the reframing of the manuscript, removal of the pilot, and the addition of the information about where samples were not available makes the story much clearer and cleaner. I also really appreciate the author's transparency on their methods including the submission of their R script assessing read depth. This was incredibly informative for my understanding of their BCR sequencing methods. Having clarified this I have one outstanding concern that I think is required for understanding their results and necessitates a reanalysis and several more minor suggestions to strengthen the clarity of the manuscript. I think this is a really interesting piece of work and am excited for it to contribute to our knowledge but would like to ensure that the results are as accurate as possible given experimental constraints and that they are not over interpreted.

Major concern: Thank you so much for including the R code you used to test the sensitivity of Alakazam to read depth. I now can better articulate my concern. In figure 3c it is evident that your sequencing depth, especially for mesenteric lymph nodes, but also for the spleens is very, very low. A MiSeq can theoretically produce 44-50 million reads passing filter -- a back of the envelope calculation using your highest read count (~2000) for every individual and sample is 0.3% of the sequencing capacity of a MiSeq run even if you grouped all the samples on a single run. Especially for splenic samples, I'd expect at least tens of thousands of reads if not more. This is very very low and makes me wonder about what went wrong in the library preparation and/or sequencing compared to the NP-Ficoll and NP-CGG experiments when the same protocol was used. I do not want to further delay the publication of this work as I know the authors have worked long and hard on this manuscript, however, if there is remaining amplicon library/ cDNA, etc. it would be very valuable to remake libraries and/or resequence to increase sampling depth. I think this is the biggest challenge of making inferences from this data because the data are inherently biased for assessment of diversity.

For reference, the alignment free method you cite for your BCR identification was tested on roughly 30K samples per distribution and was tested on datasets of equal size (aka "sampling depth"). It is here you need to subsample to the lowest number of reads -- in the clone identification -- not alakazam -- to determine whether your BCR diversity findings are accurate. You are feeding clone data from this initial assessment into alakazam and so the underlying distributions of clone calls are already biased. I think this is why the diversity estimates are so strongly (at least visually) correlated with the read counts. I bet if you plotted each individuals calculated diversity metric against its read depth there'd be a strong correlation. At minimum, I would like to see the impact of subsampling at the alignment free step -- e.g. subsampling your individual sequences to whatever the lowest sequence number is before sticking it in the python script and then doing the Hills numbers in R -- and a graph of read depth against diversity.

Minor comments:

Lines 90-116 and Figure 1: It's odd to me that you refer to parts of figure 1 in the text out of order A then D then B then C. I don't know if there's a way to rearrange the figure to flow with the text but it might be worth thinking about. Similarly in Figure 1 E you discuss IgM first in the text so maybe put that on top? 

Also in Figure 1E, it's hard to see the overlap in the IgG NP-Ficoll graph. Given that the y axes are not the same in each graph I'd just zoom in so it's clearer and there's less unnecessary white space. 

Figure 1F: I think the mouse and bat colors are switched. (Text reports the opposite findings...)

Lines 113-115: You say that mice couldn't displace the B1-8 antibody on day 56 but in your figure there's a significant difference.

Line 213: Be careful with how you're using "clonal diversity". In line 208 you use "high clonal diversity" and in 213 you use "low clonal diversity" to describe the same phenomenon. My guess is in 213 you want to use "fewer dominant clones". 

230-231: The two uses of "counterintuitive" is a little awkward. I'd remove the introductory phrase.

Other suggestions:

I saw in the reviewer comments your note about the lymph nodes. I think this is a great contribution to the field about how to find them, etc. and I think it'd be a great note for an anatomy journal or the like!

---

## [Editor Report · Decision Letter 3]

13 Aug 2024

Dear Dan,

Thank you for the submission of your revised Short Reports "Bats generate lower affinity, but higher diversity antibody responses compared to mice, an effect that can be manipulated with diet" for publication in PLOS Biology. On behalf of my colleagues and the Academic Editor, Daniel Streicker, I am pleased to say that we can in principle accept your manuscript for publication, provided you address any remaining formatting and reporting issues. These will be detailed in an email you should receive within 2-3 business days from our colleagues in the journal operations team; no action is required from you until then. Please note that we will not be able to formally accept your manuscript and schedule it for publication until you have completed any requested changes.

IMPORTANT: We routinely suggest changes to titles to ensure maximum accessibility for a broad, non-specialist readership, and to ensure they reflect the contents of the paper. In this case, we would suggest a minor edit to the title, as follows. Please ensure you change both the manuscript file and the online submission system, as they need to match for final acceptance: "Bats generate lower affinity but higher diversity antibody responses than those of mice, but pathogen-binding capacity increases if protein is restricted in their diet". 

I have asked my colleagues to include this request alongside their own.

PRESS

Sincerely, 

Melissa

Melissa Vazquez Hernandez, Ph.D., Ph.D.

Associate Editor

PLOS Biology
